# Biochemical and Initial Structural Characterization of the Monocot Chimeric Jacalin OsJAC1

**DOI:** 10.3390/ijms22115639

**Published:** 2021-05-26

**Authors:** Nikolai Huwa, Oliver H. Weiergräber, Christian Kirsch, Ulrich Schaffrath, Thomas Classen

**Affiliations:** 1Institute for Bio- and Geosciences 1: Bioorganic Chemistry, Forschungszentrum Jülich, 52425 Jülich, Germany; n.huwa@fz-juelich.de; 2Institute of Biological Information Processing 7: Structural Biochemistry, and JuStruct, Forschungszentrum Jülich, 52425 Jülich, Germany; o.h.weiergraeber@fz-juelich.de; 3Institute for Biology III, Department of Plant Physiology, RWTH Aachen University, 52056 Aachen, Germany; christian.kirsch1@rwth-aachen.de (C.K.); schaffrath@bio3.rwth-aachen.de (U.S.)

**Keywords:** monocot chimeric jacalin, lectin, OsJAC1, dirigent domain, melting point, carbohydrate-binding

## Abstract

The monocot chimeric jacalin OsJAC1 from *Oryza* *sativa* consists of a dirigent and a jacalin-related lectin domain. The corresponding gene is expressed in response to different abiotic and biotic stimuli. However, there is a lack of knowledge about the basic function of the individual domains and their contribution to the physiological role of the entire protein. In this study, we have established a heterologous expression in *Escherichia* *coli* with high yields for the full-length protein OsJAC1 as well as its individual domains. Our findings showed that the secondary structure of both domains is dominated by β-strand elements. Under reducing conditions, the native protein displayed clearly visible transition points of thermal unfolding at 59 and 85 °C, which could be attributed to the lectin and the dirigent domain, respectively. Our study identified a single carbohydrate-binding site for each domain with different specificities towards mannose and glucose (jacalin domain), and galactose moieties (dirigent domain), respectively. The recognition of different carbohydrates might explain the ability of OsJAC1 to respond to different abiotic and biotic factors. This is the first report of specific carbohydrate-binding activity of a DIR domain, shedding new light on its function in the context of this monocot chimeric jacalin.

## 1. Introduction

Lectins are found in all living organisms and viruses as soluble or membrane-bound proteins and occur both as single-domain polypeptides and as part of multi-domain arrangements [1,2]. They represent a heterogeneous group of proteins with the ability to selectively recognize and reversibly bind carbohydrates (oligo- or polymers) and glycoconjugates (glycoproteins or glycolipids) [3]. This binding specificity is key to mediate a variety of biological processes. For example, plant lectins are involved in different signaling pathways such as during development [4], in the perception of environmental signals like increased salinity [5,6], or in defense against pathogens [7]. Lectin domains can be found as part of chimeric proteins where they are fused to homologous modules or to other protein domains. More details about the different domains and functions of chimeric lectins in plants can be found in recent reviews and the references therein [8,9]. One subgroup of chimeric lectins containing a jacalin-related lectin (JRL) and a dirigent (DIR) domain is solely present in monocot plants, and are thus described as monocot chimeric jacalins [8]. JRLs are predicted to bind mannose-containing oligosaccharides and DIR-proteins might function in stereoselective coupling of monolignols. The detailed residue range of the full-length protein and its two domains is provided in Table 1.

The protein OsJAC1 from rice (*Oryza sativa*) is an example of this architecture. It consists of a JRL domain in its C-terminal region and a DIR domain in its N-terminal part [10]. OsJAC1 plays an important role in a variety of biochemical responses, including e.g., resistance against pathogens [11], cellular regulation in growth and development [12], and response to DNA damage after gamma irradiation [13].

Despite a basic biochemical characterization of OsJAC1, there is a lack of knowledge about the interplay of the two domains and how this contributes to the activity of the entire protein. Here, we report on an interaction study with the isolated JRL domain protein fused to a glutathione S-transferase (GST) tag. We determined that the domain binds selectively to mannose, thus substantiating that OsJAC1 is a mannose-binding lectin [10]. Furthermore, we investigated and reevaluated the interaction of OsJAC1 with other carbohydrates, which led to the first reported specific carbohydrate-binding activity of a DIR domain.

## 2. Results

### 2.1. Heterologous Expression and Isolation of the OsJAC1 Protein from Oryza sativa and Its JRL and DIR Domains

Since the monocot chimeric jacalin OsJAC1 has been localized in the cytosol and is not secreted [11], it was assumed that the protein is not modified by glycosylation and can be heterologously expressed in *E. coli.* After screening of different expression conditions, OsJAC1 and its two separate domains JRL and DIR were produced using a cold expression protocol (for 3.5 d at 10 °C) as detailed in Table 1.

For the first time, this approach enabled the successful heterologous production of the chimeric lectin OsJAC1 and its two domains in *E. coli* BL21(DE3). The His_6_-tagged proteins were isolated by immobilized metal-chelate affinity chromatography (IMAC) and further analyzed by size-exclusion chromatography (SEC) as depicted in Figure 1.

The purity of the recombinant OsJAC1 after IMAC isolation was 95% or better as estimated by SDS–PAGE profiles (see Figure 1a). The subsequent SEC run did not yield significant improvement of purity. Similar results were achieved for the dirigent and lectin domain proteins. The band at 40 kDa in Figure 1b was identified as the DIR domain protein by MALDI-TOF analysis. The double band at 20 kDa was also confirmed as the DIR domain protein and might have been caused by the formation of disulfide bridge during the SDS-PAGE sample preparation, the mass spectrometry could not rule out proteolysis. However, the double bands in the SDS-PAGE were avoided with a different sample preparation approach (see Appendix A).

After isolation of the full-length OsJAC1 and the DIR domain, a light yellow color of the protein solution was observed, whereas the JLR domain preparation appeared colorless.

### 2.2. Properties of the Chimeric Lectin OsJAC1—Initial Structural Characterization

#### 2.2.1. Investigation of Oligomericity

Neither three-dimensional structures nor information on the quaternary arrangement of OsJAC1 have, to our knowledge, been published to date. In order to get insights into the oligomericity of OsJAC1, we applied SEC and dynamic light scattering (DLS, Table 2). Figure 2 shows the SEC elution profiles of the chimeric lectin OsJAC1 and the isolated DIR and JRL domains.

A slight preceding shoulder can be seen in the elution profile of OsJAC1 (blue curve, Figure 2). A peak deconvolution analysis (semi-automatic peak fitting; OriginPro 2019) highlights a possible second signal, which overlaps with the main signal. The apparent mass of the secondary peak (OsJAC1_P2) corresponds to 30 kDa, which is twice the molecular weight associated with the main peak (OsJAC1_P1). It is important to note, however, that both calculated molecular weights are below the theoretical mass of 35.5 kDa. This discrepancy might be explained by a atypical running behavior due to the non-globular shape of OsJAC1 or by interaction of the protein with the column matrix, which would also lead to delayed elution. All fractions indicated in Figure 2 were investigated by SDS-PAGE (see Figure 1a) and consistently contained only the OsJAC1 protein. No difference in the elution profile or retention time was determined under reducing condition (for corresponding chromatograms refer to Appendix A).

The SEC chromatogram of the DIR domain protein showed two distinct peaks with identical composition of all fractions, as judged by SDS-PAGE (see Figure 1b). This indicates two oligomeric species. The calcultated masses of the two signals are 50 and 107 kDa, respectively, which is about twice and four times the theoretical mass of 20.9 kDa, respectively. These calculated masses thus suggest homodimeric and -tetrameric quaternary structures. Finally, the JRL domain protein showed a monodisperse peak with a calculated mass of 5.2 kDa, which is about 3.5 times lower than the theoretical mass of 18.4 kDa. Therefore, it could be speculated that properties determined by the JRL domain had the highest impact on the elution behavior of the full-length OsJAC1 protein.

Dynamic light scattering (DLS) measurements were performed in order to determine the influence of different buffer conditions on the size distrubution profile of the chimeric lectin OsJAC1 (see Table 2).

These results indicate different size distributions of OsJAC1 depending on the buffer conditions. A low buffer concentration with no additives (15 mm TRIS buffer) resulted in two distinct particle populations. The two species differ in their apparent hydrodynamic radius by an order of magnitude. The large particles in the MDa range indicated aggregation of the protein, which was aggravated by applying denaturing conditions (CaCl_2_), which led to a broad size dispersity.

Conversely, the application of DTT (dithiothreitol; reducing conditions), lactose, or a non-chaotropic salt like MgSO_4_ led to mono-dispersity. Lactose has been added because the binding assay (see below) identified this disaccharide as a ligand.

However, all determined molecular weights are at least four times higher than the theoretical mass of 35.7 kDa. Since the determination of apparent hydrodynamic radii from diffusion coefficients is affected by deviations of particle shape from sphericity, these results might indicate that OsJAC1 is not a globularly shaped protein.

Notably, our findings *in planta* also indicate oligomerization of OsJAC1. Figure 3 shows a temperature dependent blot assay of a control protein (green fluorescent protein; *GFP*) and OsJAC1 as fusion protein with GFP after protein isolation from transgenic barley plants overexpressing either *GFP* or *OsJAC1-GFP*.

The GFP protein isolated from transgenic barley plants was detected on Western blots in a monomeric state at 26 kDa, independent of pretreatment with different temperatures (Figure 3; left panel). Contrarily, the fusion protein OsJAC1-GFP displayed two oligomeric states at approximately 60 and >155 kDa, respectively, at temperatures below 75 °C, while at higher temperature, only the monomeric state at 60 kDa could be detected (Figure 3; right panel).

#### 2.2.2. Melting Point (T_m_)

In order to gain information on the quality and stability of the recombinantly expressed proteins, melting curves were recorded via differential scanning fluorimetry (DSF). The thermal shift assay is based on monitoring of the intrinsic fluorescence signal of proteins after emission at 350/330 nm. The latter predominantly originates from tryptophan residues and is sensitive to their environment, with a red shift indicating transition into a more polar setting. Folded proteins thus typically exhibit emission maxima (λ_max,em_) at about 330 nm, while peaks around 350 nm are found for unfolded states [14]. The ratio of the signals at these two wavelengths is plotted against the temperature to determine the melting point (T_m_) of a protein. The melting curves of OsJAC1 and its isolated domains (independent and mixed) under both reducing and oxidizing condition can be seen in Figure 4 and a summary of the results is given in Table 3.

The thermal unfolding profiles differ in each case. The DIR domain protein (grey) showed the overall lowest initial ratio as well as the highest Δ ratio, which is the ratio difference between the beginning and the end of the melting curve. Moreover, it is quite insensitive to the redox potential of the medium, only a slight decrease of the melting point under reducing conditions can be observed.

In contrast to the behavior of the DIR domain, the melting curve of the JRL single domain protein showed a very high initial ratio compared to the other curves. With increasing temperature, the value smoothly decreased, until a distinct thermal unfolding transition occurred. A comparison of the saccharide-binding sites of the JRL homolog Banlec from the banana fruit (*Musa acuminata*) [15] and the JRL domain of OsJAC1 reveals the presence of a tryptophan residue in position W60 of the latter (see Appendix A). This tryptophan might be exposed to the surface and could thus explain the overall high initial fluorescence intensity ratio of the JRL domain protein. In fact, the distinct progression of the melting curves and the prominent transition at about 59 °C were observed for all samples containing the JRL domain (JRL itself, full-length OsJAC1, JRL + DIR). Furthermore, due to the presence of one cysteine within the JRL single domain protein, only intermolecular disulfide bonds within homodimers may be formed under non-reducing conditions. This putative disulfide bond between JRL domains indeed appears to highly increase the stability of the isolated domain since its apparent melting temperature shifts by 16 K compared to the reduced form. In the absence of the disulfide bond, the first inflection point (at T_m1_) is associated with substantial changes to the environment of tryptophan residues, possibly indicating denaturation of the JRL domain or a subunit thereof, whereas under non-reducing conditions, this transition is barely visible and the majority of the ratio shift occurs at T_m2_. Since the chimeric lectin OsJAC1 is natively present in a reducing environment (cytosol), it can be assumed that the two transition points reflect the innate thermal unfolding profile.

As it was proven that the separated domains interact in vivo [11], 1:1 mixtures of the individually expressed domains were tested as well. Under both non-reducing and reducing conditions, the melting temperatures observed for the mixture largely coincided with those of the individual domains, whereas in full-length, OsJAC1 T_m2_ is shifted upwards by approx. 8 K (Table 3). Since this transition appears to be dominated by the DIR domain, our observations suggest that the physical linkage of the two modules entails a stabilizing effect mostly on the DIR domain.

#### 2.2.3. Circular Dichroism (CD) Spectroscopy

In order to get first insights into the secondary structure elements of OsJAC1 and its two domains (JRL & DIR), circular dichroism spectroscopy was applied under reducing conditions. Figure 5 displays the smoothed CD-spectra of the three proteins (full protein, DIR and JRL domain proteins) and the deduced secondary structure composition (%).

The webtool BeStSel was used for the secondary structure evaluation [17]. In general, the similarities in secondary structure content between the two single domain proteins and the full-length protein are very high. The prevailing structure elements are antiparallel beta-sheets as can be seen in Figure 5. The DIR domain protein is suggested to additionally contain a few residues in parallel beta-sheets. However, this information should be treated with caution since the fraction is very low and not detected at all for the full OsJAC1 protein. A conformational change due to the interaction between the two domains in the chimeric lectin OsJAC1 cannot be identified from these data. Table 4 shows a comparison of the CD-derived secondary structure estimates for the two OsJAC1 domains with the composition of selected homologs, where three-dimensional structure information is available.

Both the dirigent domain and the jacalin domain are dominated by their β-strand-content and contain no α-helices. The DIR domain of OsJAC1 exhibits a secondary structure composition that is remarkably similar to its homologs, while the content of the various JRL homologs appears relatively more diverse.

### 2.3. Interaction Partners of the Lectin Domain

In general, interactions with binding partners such as small-molecule ligands tend to increase the stability of a protein [22]. This relation was used to screen for potential sugar interaction partners for the JRL domain of the chimeric lectin OsJAC1 via differential scanning fluorimetry; an increase of the melting point was used as an indication of ligand binding. As shown above, there are two inflection points within the melting curve of OsJAC1 under reducing conditions (see Figure 4), and for saccharide-binding studies, we initially focused on the first one (T_m1_), which was attributed to the JRL domain. Notably, T_m1_ for OsJAC1 varied between 59 and 61 °C for individual isolation batches, whereas the shifts induced by the presence of sugars (∆T_m1_) remained the same. The observed variation in the first inflection point was therefore attributed to minimal changes of buffer composition between batches, and the evaluation of ligand effects was based solely on ∆T_m1_ (T_m1,sugar_ − T_m1,control_) instead of raw T_m1_. Figure 6 shows the determined shifts of inflection points (induced by different selected mono- and disaccharides) for both full-length OsJAC1 and the separated JRL domain in a stoichiometric mixture with the DIR domain (JRL + DIR). The inflection point of the single JRL domain protein was in most cases undetectable under reducing conditions and therefore not suitable for the screening of possible sugar interaction partners. However, by appling a mixture of the single JRL and the DIR domain protein, the detectability of the JRL inflection point was improved and hence suitable for the screening.

At first, we tested different monosaccharides with a focus on mannose, since Jiang et al. reported the recognition and specific binding of mannose by the JRL domain [10]. A slight increase of ∆T_m1_ by 1 K was observed in the presence of mannose and glucose for both OsJAC1 and the single domain JRL protein. In a second sugar screening trial, different disaccharides were tested, such as *N*,*N*′-diacetylchitobiose (chitobiose; component of the fungal cell wall) or glucose-containing saccharides (e.g., cellobiose or laminaribiose). The disaccharides 1,2-α-mannobiose and especially laminaribiose were able to increase the T_m1_ by 2–3 K compared to the control. The data also showed that the JRL domain protein is able to distinguish between different glucosyl disaccharides, with the β1,3-linked laminaribiose showing the largest effect. These results emphasize the suitability of the differential scanning fluorimetry for interaction studies with a variety of sugars.

Besides the ligand-related effects on T_m1_ (outlined above), we also observed increases in the second melting temperature (T_m2_ = 85 °C) of OsJAC1 upon addition of sugars (see Figure 7). Data shown in Figure 4 suggest that this transition is exclusively due to the DIR domain.

Of the different monosaccharides tested, galactose showed the highest increase in T_m2_ of up to 2 K for OsJAC1 and 3 K for the DIR domain. Even larger effects, up to 6 K for OsJAC1 and 10 K for the DIR domain, were observed with galactose-containing disaccharides. This sugar-related increase in stability of the DIR domain protein was unexpected. To test whether elevation of the melting point was due to an unspecific interaction of galactose, lysozyme and serum albumin were used as negative controls (results in the Appendix A). The tested sugars (e.g., galactose and lactose) had a positive impact on T_M2_ of the DIR domain and OsJAC1 proteins, however showed no effect on the controls. A cluster analysis with k-means of all data confirmed this observation, i.e., was able to distinguish between saccharides which contain galactose and those which do not, as shown in Figure 7.

The influence of the oxidative state on the binding ability was tested for OsJAC1 and its single domain proteins (results in Appendix A). Both single domain proteins had the same sugar response under oxidizing and reducing conditions. Consistent with this observation, T_m2_ of full-length OsJAC1 and its response to sugars were largely unaffected by the redox condition. In contrast, the isolated JRL domain responses are not straightforward to compare since the characteristic transition at T_m1_ of the full-length protein OsJAC1 is barely detectable under non-reducing conditions.

Near-UV CD spectroscopy was applied to further probe the sugar interaction (spectra see Appendix A). The binding partners with the most prominent effects in our DSF experiments, i.e., galactobiose (DIR domain) and laminaribiose (JRL domain), caused major shifts in the CD spectra. Both methods (near-UV CD and DSF) thus support the notion that the two domains of OsJAC1 possess different specific saccharide-binding sites with affinities towards galactose and glucose/mannose moieties, respectively.

## 3. Discussion

The present study was conducted to collect in-depth information about biochemical properties of the monocot chimeric jacalin OsJAC1, which was shown in previous studies to accumulate focally around fungal infection sites in transgenic barley [11]. In the long run, this should reveal a potential binding partner of OsJAC1 either derived from pathogens or from the plant in response to its infection. Additionally, this might enable to harness this knowledge for antifungal treatments. As the first milestone, we established the heterologous production of OsJAC1 and its two single domain proteins in *E. coli*, which was not reported before. The soluble production of the proteins was only achieved under cold expression conditions. Throughout the low-temperature incubation, the expression rate was reduced, which seemed to be beneficial for proper folding of the proteins.

Known JRL and dirigent proteins consist of β-sheet structure elements, exclusively [21,23]. JRLs form a β-prism I fold architecture [24] and dirigent proteins are known for their β-barrel structures [21,25]. The CD measurements (Figure 5) of the two separated domain proteins verified the rich β-sheet content. Accordingly, it can be assumed that both domains share the characteristic molecular structure of their respective families. Figure 8 gives an overview of the results in a model for OsJAC1.

The melting point of the DIR domain is quite insensitive to the redox potential of the medium, only a slight decrease was observed under reducing conditions (Figure 4). Hence, there are no indication that the DIR domain is able to form disulfide bonds. In case of the JRL domain protein, we demonstrated that it is less stable under reducing conditions, which can lead to the removal of an intermolecular disulfide bond within a JRL homodimer. However, since the chimeric lectin OsJAC1 is natively present in a reducing environment, the cytosol, it can be assumed that the formation of disulfide bonds is disfavored [26].

The present results showed that only the DIR domain experiences a stabilization (by up to 8 K) in the full-length protein compared to the separated single domain protein (Figure 4). This suggests a possible interaction with a structurally relevant subunit, although the majority of the JRL domain is already unfolded, which has been validated by measurements of pretreated heat denatured samples (data not shown).

The SEC-deduced masses of the two overlapping OsJAC1 signals (14.8 and 30 kDa, Figure 2) are lower than the theoretical mass of the monomer (35.57 kDa), independent of their redox state. The column material consists of a dextran matrix (glucosyl polysaccharide) that is covalently bound to a cross-linked agarose. As shown in the sugar interaction study (Figure 6), the JRL domain can recognize specific glycosyl moieties, which might lead to an interaction with the column matrix and subsequent increase in retention time. Nevertheless, the results of the DLS and trials *in planta* suggest the presence of two dynamically formed oligomeric OsJAC1 species under the applied conditions (Figure 3). The formation of disulfide-linked homodimers has been reported for a JRL homolog in *Pteria penguin* pearl shell [27] and is assumed to also occur in a JRL homolog from fungi [28]. To the best of our knowledge, the JRL domain of OsJAC1 would be the first identified JRL representative from the plant kingdom that can form disulfide linkages within homodimers under oxidizing conditions. Due to the possibility of inter-subunit disulfide bond formation, the quaternary structure of OsJAC1 may represent an asymmetric dimer configuration as described for the JRL homologs of pearl shell (*Pteria penguin*) [27] or of banana fruit (*Musa acuminata*) [15]. In case of the JRL domain of OsJAC1, the linkage of the subunits must be close to the N-terminal boundary (C184). This kind of tail-to-tail configuration can be also found in other JRL homologs which typically form tetramers [29,30]. Interestingly, SEC elution profiles of the separated DIR domain also indicate the presence of a dimeric and a less populated tetrameric state (Figure 2).

It has been previously reported that JRL domains of chimeric lectins showed alterations of their carbohydrate-binding specificity when separated from their respective DIR domains [31]. In contrast, our results on OsJAC1 show that the carbohydrate-binding specificity of the JRL single domain protein was unchanged in the absence of the DIR domain. The interaction study revealed that the presence of disaccharides containing galactose, mannose, and glucose increased the stability to a greater extent than the corresponding monosaccharides. The melting points assigned to the two domains increase in the presence of sugars independent of whether the domains were expressed separately or within OsJAC1. The DIR domain protein specifically recognizes sugars containing galactose, but not *N*-acetyl-d-galactosamine (GalNAc) (Figure 7). The binding region related to the JRL domain showed affinity towards glucose and mannose (Figure 6). Mannose and glucose have a similar contact profile by which lectins recognize the equatorial position of the 4-hydroxyl group, in contrast to the axial 4-hydroxyl group in galactose [32]. Indeed, classical JRLs are divided into galactose- and mannose-specific lectins [33,34]. There is increasing evidence, however, that the monocot chimeric jacalins exhibit a higher diversity in their binding patterns than their classical counterparts [8]. For example, BGAF from *Sorghum bicolor* is highly selective towards GalNac [35], whereas maize BGAF (H95 Inbred Line) shows a recognition pattern similar to OsJAC1, thus responding to galactose and mannose [36]. Recently, Ma et al. showed that homologs of OsJAC1 in wheat are able to interact with galactose and mannose [37].

Classical dirigent proteins are predicted to be involved in lignan synthesis and it is assumed that the monolignol-derived educts are bound and positioned by the protein for a stereoselective radical reaction [38]. In the context of chimeric jacalins, however, DIR domains may exert modified or additional roles. It was reported, for instance, that the chimeric lectin from wheat TaJA1 did not lead to an increase of lignan content [39] and in BGAF both domains were needed for aggregation activity [36]. In this study, we demonstrated that the DIR domain of OsJAC1 showed a broad sugar interaction capability with a preference for different galactose-containing disaccharides. Therefore, the role of the DIR domain in this monocot chimeric jacalin may need to be reconsidered. The ability of the OsJAC1 DIR domain to specifically bind different sugar moieties might hint at its participation in recognizing pathogen-associated molecular patterns. Whether the discrete specificity introduced by the DIR domain assists in localization of the entire protein or is involved in modulation of the dirigent activity, will be subject of future studies.

## 4. Materials and Methods

### 4.1. Materials

Buffers were prepared or diluted with ddH_2_O, if not otherwise indicated. d(+)-galactose was purchased from AppliChem GmbH (Darmstadt, Germany). α-d-(+)-glucose, d-(+)-melibiose monohydrate, *N*-acetylneuraminic acid, β-d-gentiobiose, and lactose were purchased from Carl Roth GmbH + Co. KG (Karlsruhe, Germany). *N*,*N*′-diacetylchitobiose was purchased from Carbosynth Limited (Berkshire, UK). d-sorbitol, d-(+)-xylose, l-rhamnose monohydrate, and *N*-acetyl-d-galactosamine were purchased from Merck KGaA (Darmstadt, Germany). 1,4-β-d-galactobiose, galactan, and laminaribiose were purchased from Megazyme (Bray, Ireland). Sucrose and dithiothreitol (DTT) were purchased from Fisher Chemical (Schwerte, Germany). d-(+)-maltose monohydrate was purchased from TCI Deutschland GmbH (Eschborn, Germany). d-(+)-mannose was purchased from Thermo Fisher Scientific GmbH.—Acros Organics (Schwerte, Germany). *N*-acetyl-β-d-glucosamin was purchased from GLYCON Biochemicals GmbH (Luckenwalde, Germany). 2α-mannobiose was purchased from Dextra Laboratories Ltd. (Reading, UK).

### 4.2. Western Blot Using Transgenic Plants Tissue

Transgenic barley plants overexpressing GFP were kindly provided by Prof. Dr. Ralph Hueckelhoven (TU Munich). OsJAC1-GFP overexpressing barley plants are described already in Weidenbach et al. (2016). Plant seeds were pregerminated on wet filter paper overnight by room temperature. The following day, germinated seeds were sowed in ED73 soil (Balster Einheitserde GmbH, Fröndenberg, Germany) and incubated at 16 h daylight (approx. 200 µmol m^−2^ s^−1^), 18 °C, and 65% relative humidity. Primary leaves were harvested seven days after germination and frozen in liquid nitrogen. For protein isolation, 150 mg of ground frozen tissue were transferred into a 1.5 mL reaction tube. Then, 400 µL of chilled extraction buffer (50 mM Tris-HCl pH 8.0, 1 mM EDTA, 10 mM DTT, and 1/1000 vol. phenylmethylsulfonyl fluoride (PMSF) (30 mg/mL stocksolution, dissolved in ethanol) were added, mixed, and kept on ice. Samples were centrifuged for 20 min at 16.000× *g* and 4 °C and the supernatant was transferred into a new tube. This step was repeated once more with the supernatant. Then, 25 µL of the supernatant was mixed with 2× protein gel loading buffer (NuPAGE Sample Reducing Agent (10X) and NuPAGE™ LDS Sample Buffer (4X), Thermo Fisher Scientific GmbH—Invitrogen, Scherte, Germany) and incubated for 10 min at different temperatures between 60 and 95 °C. Samples were loaded and separated under denaturing conditions on a Bis-Tris gel using MES running buffer (50 mM MES, 50 mM Tris, 1 mM EDTA, 0.1% (*w*/*v*) SDS). Afterwards, a Western blot was performed using primary Anti-GFP antibody (mouse, Roche Deutschland Holding GmbH, Mannheim, Germany) and secondary Anti-mouse IgG, HRP-linked antibody (Cell Signaling Technology, Danvers, MA, USA).

### 4.3. Construct and Cloning

All genes were present in pET-15b(+) vectors encoding an N-terminal His_6_-tag and a thrombin cleavage site. The OsJAC1 gene was additionally cloned into pColdIV (Takara Bio Inc., Kusatsu City, Japan) by Gibson assembly. The amino acid sequences were as follows: 

JRL domain:

MGSSHHHHHHSSGLVPRGSHMLEQCPVTKIGPWGSSHEGTVQDITESPKRLESITLY HGWSVDSISFTYLDHAGEKHKAGPWGGPGGDPIMIEFGSSEFLKEVSGTFGPYEGSTVITSINFITNKQTYGPFGRQEGTPFSVPAQNNSSIVGFFGRSGKYINAVGVYVQPI

DIR domain:

MGSSHHHHHHSSGLVPRGSHMLESKLQITPCGMLVQGNQINFTKLYLHHTPAGPEQNQSAVTSNDKK*I*GLGCIVVNNWSVYDGIGSDAKLVAYAKGLHVFAGAWHNSFSLVFEDERLKGSTLQVMGLIVEEGDWAIVGGTGQFAMATGVILKKMQEQKQYGNIIELTIHGFCPLLKGS

The marked amino acid (*) in the artificial separated DIR domain protein is a point mutation from threonine to isoleucine.

OsJAC1:

MGSSHHHHHHSSGLVPRGSHMLEMADPSKLQITPCGMLVQGNQINFTKLYLHHTPAGPEQNQSAVTSNDKKTGLGCIVVNNWSVYDGIGSDAKLVAYAKGLHVFAGAWHNSFSLVFEDERLKGSTLQVMGLIVEEGDWAIVGGTGQFAMATGVILKKMQEQKQYGNIIELTIHGFCPLLKGSQCPVTKIGPWGSSHEGTVQDITESPKRLESITLYHGWSVDSISFTYLDHAGEKHKAGPWGGPGGDPIMIEFGSSEFLKEVSGTFGPYEGSTVITSINFITNKQTYGPFGRQEGTPFSVPAQNNSSIVGFFGRSGKYINAVGVYVQPI

### 4.4. Heterologous Expression of OsJAC1 and Its DIR and JRL Domains

Large scale heterologous expression of *OsJAC1*, *JRL*, and *DIR* domain was achieved in *E. coli* BL21(DE3). The general approach was as follows: 5 L baffled flask with 1 L LB medium, 100 µg/mL ampicillin and 2% ethanol were inoculated with 10 mL preculture, incubated at 37 °C and 200 rpm (New Brunswick Innova 42 Incubator Shaker, Eppendorf, Hamburg, Germany). After an OD_600_ value between 0.4–0.6 had been reached, a cold shock was performed, where the flasks were incubated on ice for 0.5 h. Prior to the induction with 100 µm IPTG, the cells were additionally incubated at 10 °C for 0.5 h and 200 rpm. After induction, the cells were incubated for 3.5 d at 10 °C. The cells were harvested by centrifugation (15 min, 3000× *g*, 4 °C; Sorvall RC 6 Plus Centrifuge, Thermo Scientific, Schwerte, Germany) and finally stored at −20 °C.

### 4.5. Isolation of OsJAC1 and the Single Domains (DIR and JRL)

Approximately 5 g of cell pellet were used for each isolation batch and suspended in 10 mL buffer A (50 mm TRIS-HCl, 500 mm NaCl, 20 mm imidazole at pH 7.4). Cells were lysed by sonicator Sonoplus HD220 for 5 × 10 cycles at 37% power, and 5 min in total (BANDELIN electronic GmbH & Co. KG, Berlin, Germany). The lysate was centrifuged (18,000× *g*, 4 °C, 30 min) and the supernatant filtered by a syringe filter (0.45 µm pores; VWR International GmbH, Darmstadt, Germany).

The proteins were isolated at room temperature using a Ni-NTA column (1 CV = 5 mL) purchased from IBA GmbH (Goettingen, Germany). First, the column was rinsed with 3 CV of water and then equilibrated with 5 CVs of Buffer A at a flow rate of 2.5 mL/min. The lysate was loaded in cycle on the column by a peristaltic pump (Pharmacia Biotech-Pfizer Deutschland GmbH, Berlin, Germany) for 0.5 h. The washing step was executed with 12 CV of buffer A and followed by a one-step increase of buffer B (50 mm TRIS-HCl, 500 mm NaCl, 250 mm imidazole at pH 7.4) in order to elute unselective bound impurities (30% buffer B for 8–12 CV). The protein was eluted by buffer B (100%) for 10 CV. Elution buffer containing the protein was exchanged by 50 mm potassium phosphate at a pH of 7.4 by PD-10 columns (Cytiva Europe GmbH, Freiburg, Germany).

For the size exclusion chromatography, was used the Cytiva (formerly GE Healthcare Life Sciences) HiLoad™ 16/600 Superdex™ 200 pg (120 cm × Ø 1.6 cm). The protein and elution buffer consisted of 50 mm KPi, 300 mm NaCl, and pH of 7.4. Prior to injection, the protein solution was filtered by a syringe filter (0.45 µm pores; VWR International GmbH, Darmstadt, Germany). Then, 1 mL protein sample (10 mg/mL) was injected and eluted with a flowrate of 1 mL/min. Fractions of 4 mL were sampled.

In the first SDS-PAGE attempt, the samples were heated at 95 °C for 15 min prior to loading. In order to avoid double bands caused by disulfide bridges, the samples were incubated at 70 °C for 10 min.

### 4.6. Protein Concentration Determination

The protein concentration was quantified via absorbance measurement at a wavelength of 280 nm (NanoDrop 2000c, ThermoFisher, Schwerte, Germany). The molar extinction coefficients of OsJAC1 (47,900 m^−1^ cm^−1^) and its two domain proteins (JRL: 30,940 m^−1^ cm^−1^; DIR: 22,460 m^−1^ cm^−1^) were determined using the online tool ProtParam.

### 4.7. MALDI-TOF-MS Analysis

The desired band was cut out of the gel by using a scalpel and transferred into a 1.5 mL reaction tube. Then, 750 µL of 30% (*v*/*v*) acetonitrile in 100 mm NH_4_HCO_3_ were added and the tube was shaken for 10 min at room temperature (RT) very gently. The supernatant was carefully discarded and these steps were repeated twice. The remaining solvent was evaporated using a vacuum centrifuge for 20 min at 37 °C. For rehydration of the gel piece, 6 µL TRIS-HCl buffer (3 mm, pH 8.8) containing trypsin (10 ng/µL) was added. The sample was centrifuged to collect the entire material and incubated for 30 min at RT. Further, 6 µL of TRIS/trypsin buffer was added and incubated over night at RT. After incubation, 10 µL ddH_2_O were added and incubated for additional 15 min. Finally, 10 µL 30% (*v*/*v*) acetonitrile with 0.2% (*v*/*v*) trifluoroacetic acid were added and incubated for 10 min. The samples were stored at −20 °C. Measurements were conducted at the Institute of Bio- and Geosciences IBG-1: Biotechnology (Forschungszentrum Jülich) using an Ultraflex III TOF/TOF mass spectrometer (Bruker Daltonics, Bremen, Germany).

### 4.8. DLS

Dynamic light scattering experiments were conducted on a SpectroSize 300 instrument (Xtal Concepts GmbH, Hamburg, Germany). All OsJAC1 samples (protein concentration 1.1 mg/mL) were centrifuged at 20,000 × *g* and 4 °C for 30 min prior to the measurements. Data were recorded in 25 successive acquisitions, each covering a 10-s interval.

### 4.9. Differential Scanning Fluorimetry

The melting curves of the proteins were determined using Prometheus NT.Plex nano DSF (NanoTemper, München, Germany). For all sugars investigated, 100 mm stock solutions were prepared and diluted (1:1) with the protein solution so that the final sugar concentration was 50 mm. As buffer, 50 mm KPi with a pH of 7.4 was used. Galactan solution was prepared as described in the manual; briefly, 100 mg was diluted in 10 mL water and heated for 20 min at 60 °C until a clear solution had formed. The final protein concentration for the assay ranged between 0.5 and 1.87 mg/mL. Samples were incubated for 0.5 h prior to the thermal shift assay. All measurements were performed in triplicates. Fluorescence was recorded with 7% excitation power, scanning a temperature range from 20 to 95 °C at a rate of 0.5 °C/min.

### 4.10. CD Spectroscopy

CD spectra were recorded on a J-1100 Spectropolarimeter (JASCO Deutschland GmbH, Pfungstadt, Germany). All samples were centrifuged at 20,000 × *g* and 4 °C for 30 min prior to the measurement.

Far-UV scans (260–190 nm, five acquisitions per sample) were recorded at a scan rate of 50 nm/min with a data pitch of 1 nm, using a 0.2-mm path length cuvette (Hellma GmbH & Co. KG, Mühlheim, Germany). Samples were provided in 10 mm KPi, pH 7.4, containing 1 mm DTT (OsJAC1: 1.52 mg/mL; DIR domain: 1.04 mg/mL; JRL domain: 2.39 mg/mL).

Near-UV scans (330–260 nm, five acquisitions per sample) were recorded at a scan rate of 20 nm/min with a data pitch of 0.5 nm, using a 1-mm path length cuvette (Hellma GmbH & Co. KG, Mühlheim, Germany). Samples were provided in 50 mm KPi, pH 7.4, containing 1 mm laminaribiose or galactobiose (OsJAC1: 7.67 mg/mL; DIR domain: 7.93 mg/mL; JRL domain: 15.58 mg/mL).

## Figures and Tables

**Figure 1 ijms-22-05639-f001:**
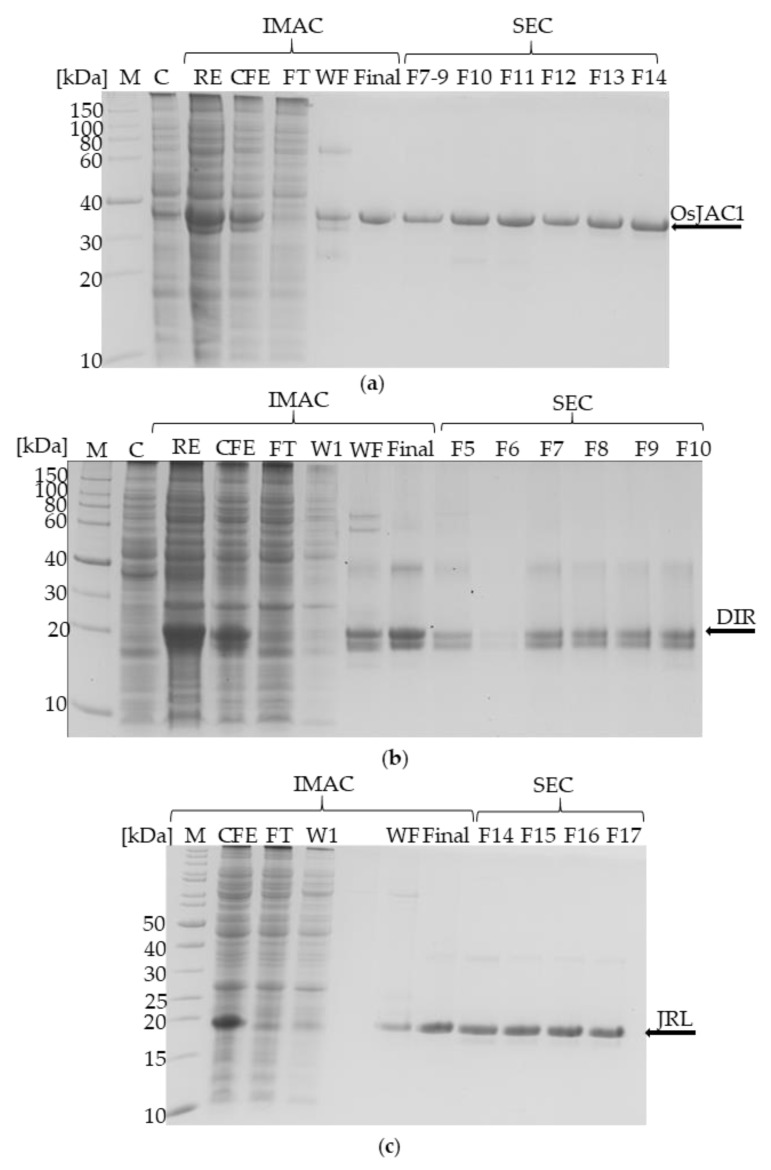
Coomassie-stained SDS-PAGE (12%) visualizing heterologous production and isolation of OsJAC1 (**a**), its DIR domain (**b**), and its JRL domain (**c**). M = marker; C = control; RE = raw extract/disrupted cell suspension; CFE = cell free extract; FT = flow through; W1 = wash fraction with 100% buffer A (20 mm imidazol); WF = wash fraction with 30% buffer B (250 mm imidazol); F# = SEC fraction (size-exclusion chromatography); Final = final solution after IMAC.

**Figure 2 ijms-22-05639-f002:**
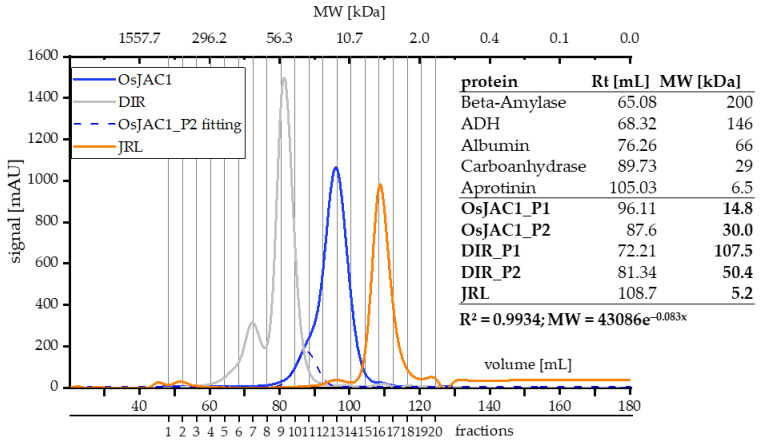
SEC chromatograms of OsJAC1-related proteins: full-length OsJAC1 (blue), JRL domain (orange), and DIR domain (grey). The inserted table lists the parameters of proteins used for calibration along with the observed retention times and deduced apparent molecular masses of OsJAC1 proteins. SEC-buffer: 50 mm KPi-buffer, pH 7.4, and 300 mm NaCl.

**Figure 3 ijms-22-05639-f003:**
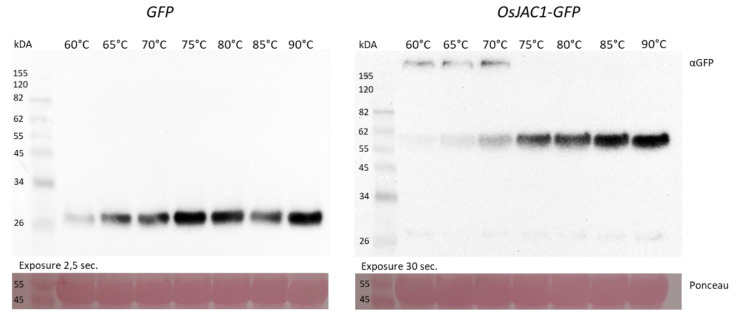
Western blots using anti-GFP antibody and proteins isolated from transgenic barley plants overexpressing either green fluorescent protein (*GFP;*
**left**) or *OsJAC1-GFP* (**right**). Proteins were incubated at different temperatures for 10 min prior to loading onto the BisTris gel using MES running buffer.

**Figure 4 ijms-22-05639-f004:**
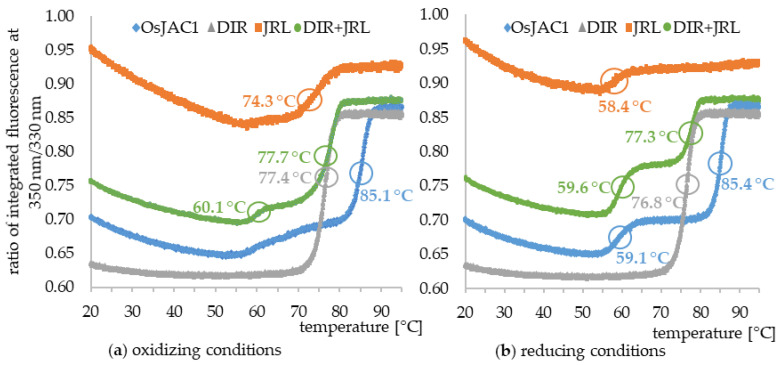
Melting curves of OsJAC1 (blue), its two separated domains JRL and DIR (orange and grey), and a stoichiometric mixture of the domains (green) in (**a**) 50 mm KPi buffer (pH 7.4), (**b**) 50 mm KPi buffer (pH 7.4), and 4 mm DTT. Inflection points of the transitions are circled and respective melting temperatures (T_m_) indicated.

**Figure 5 ijms-22-05639-f005:**
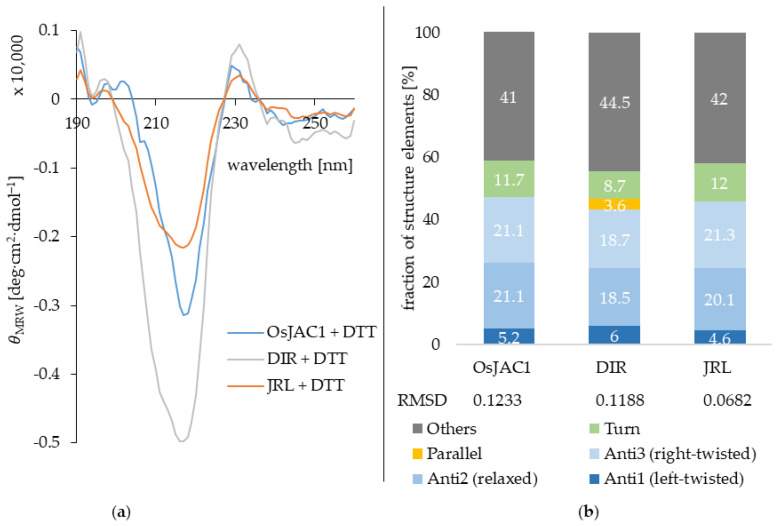
(**a**) Savitzky–Golay-smoothed [16] CD spectra of the full-length protein OsJAC1 and the single domain proteins JRL and DIR under reducing conditions (10 mm KPi buffer, 4 mm DTT, pH 7.4). (**b**) Estimated secondary structure content (%) determined by the webtool BeStSel [17].

**Figure 6 ijms-22-05639-f006:**
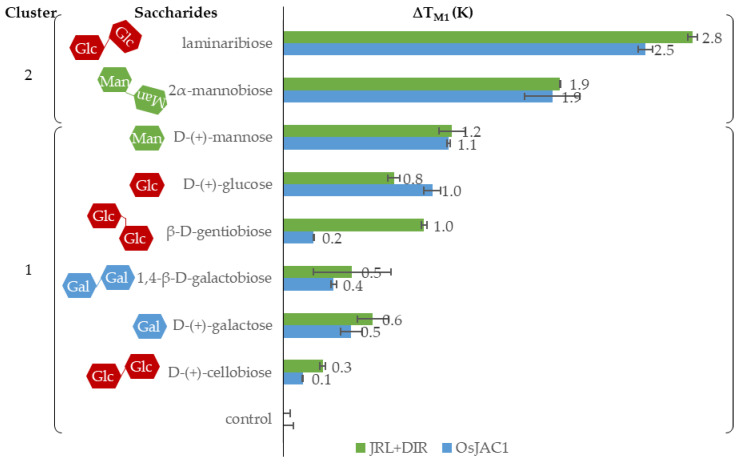
Mean T_m1_ of OsJAC1 (blue) and the JRL domain in a 1:1 mixture with the DIR domain (green; JRL + DIR) after addition of different saccharides (50 mm) under reducing conditions (4 mm DTT). Analyzed with Cluster analysis by K-Means (OriginPro 2019).

**Figure 7 ijms-22-05639-f007:**
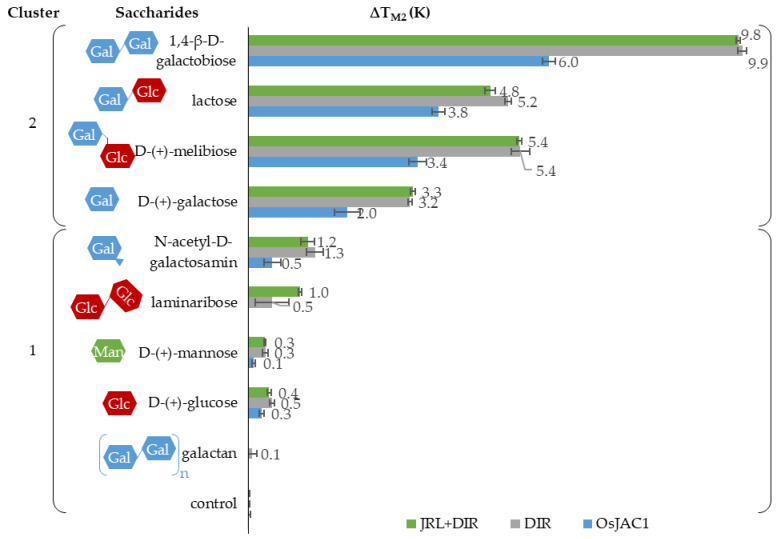
Mean ∆T_m2_ for OsJAC1, the isolated DIR domain protein and the JRL + DIR (1:1) protein mixture after addition of different saccharides (50 mm) under reducing conditions (4 mm DTT). Analyzed with cluster analysis by k-means (OriginPro 2019).

**Figure 8 ijms-22-05639-f008:**
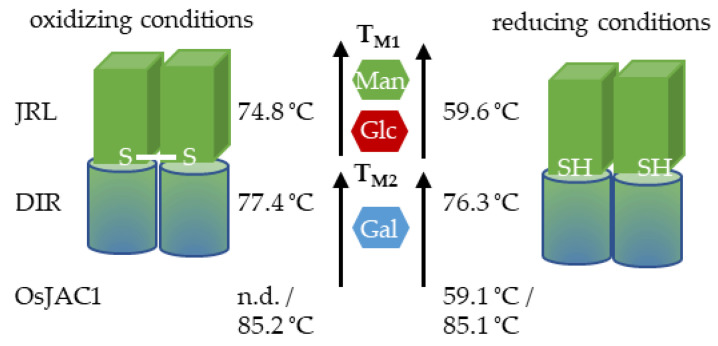
A model for OsJAC1 as a dimer. The JRL domain has one cysteine (C184) and it may form an intermolecular disulfide bond under oxidizing condition. The given temperatures are melting points for the respective domains and the full-length protein under the given conditions. Mannose and glucose are part of a putative JRL ligand and galactose may be part of a DIR ligand.

**Table 1 ijms-22-05639-t001:** Overview of the expression settings with *Escherichia coli* BL21(DE3) and exemplary results of the isolation for OsJAC1 and its two artificially separated domains (JRL and DIR) via immobilized metal-chelate affinity chromatography (IMAC).

	OsJAC1	DIR	JRL
residue range	1–329	28–182	183–329
molecular mass [kDa]	35.57	19.3	18.39
plasmid	pColdIV	pET15b(+)	pET15b(+)
expression condition	10 °C for 3.5 d	10 °C for 3.5 d	10 °C for 3.5 d
pellet size [g]	7	5.8	8.6
purity [%] ^a^	>95	>95	>95
conc. after IMAC [mg mL^−1^]	10.7	9.7	8.1
V_Final_ [mL]	3.5	3.5	3.5
appearance	light yellow color	light yellow color	colorless

^a^ as judged by SDS-PAGE.

**Table 2 ijms-22-05639-t002:** Dispersity and apparent molecular mass of OsJAC1 in relation to different buffer additives, as determined by DLS. The calculated molecular weights of the most abundant species are listed in the table and the corresponding plots can be found in the (Appendix A). All tested conditions were set up with 15 mm TRIS-HCl buffer.

Buffer Condition	Min. MW/kDa	Max. MW/MDa	No. of Distinct Species
15 mm TRIS-HCl pH 7.4	308	26.95	two
+150 mm CaCl_2_	268.8	3317	many
+50 mm lactose	187.4	-	one
+150 mm MgSO_4_	183.7	-	one
+4 mm DTT	137.5	-	one

**Table 3 ijms-22-05639-t003:** Summarized results of the thermal shift assay for OsJAC1 and proteins with the single domains.

DTT	Sample	Initial ratio ^a^	∆ ratio ^b^	T_M1_ (°C) (±)	T_M2_ (°C) (±)
−	JRL	0.95	−0.03	n.d.	74.3 ± 0.04
−	DIR	0.63	0.23	n.a.	77.4 ± 0.01
−	OsJAC1	0.70	0.17	n.d.	85.1 ± 0.03
−	JRL + DIR	0.76	0.12	60.1 ± 0.02	77.7 ± 0.02
+	JRL	0.96	−0.03	58.4 ± 0.13	n.d.
+	DIR	0.56	0.28	n.a.	76.8 ± 0.03
+	OsJAC1	0.70	0.16	59.1 ± 0.01	85.4 ± 0.01
+	JRL + DIR	0.77	0.12	59.6 ± 0.51	77.3 ± 0.05

^a^ I(350 nm)/I(330 nm); ^b^ [I(350 nm, 95 °C)/I(330 nm, 95 °C)] − I[(350 nm, 20 °C)/I(330 nm, 20 °C)]; n.d. not determinable; n.a. not available.

**Table 4 ijms-22-05639-t004:** Comparison of secondary structure compositions determined by CD spectroscopy for the OsJAC1 domain proteins and by X-ray crystallography for certain homologs [jacalins from *Morus indica* (gJRL), and from *Musa acuminata* (mJRL); dirigent proteins from *Glycyrrhiza echinate* (GePTS1), and from *Arabidopsis thaliana* (AtDIR6)].

Protein	α-Helix	β-Sheet	β-Turns	Other	Reference
OsJAC1 JRL domain	0%	46%	12%	42%	-
gJRL [18]	2%	34%	23%	39%	[18]
mJRL	0%	73% ^1^	-	27%	pdb:4PIF [19]
OsJAC1 DIR domain	1%	47%	9%	44%	-
Dirigent protein GePTS1	4%	56% ^1^	-	40%	pdb:6OOC [20]
Dirigent protein AtDIR6	0%	56% ^1^	-	44%	pdb:5LAL [21]

^1^ β-strand according to the PDBsum entry.

## Data Availability

Not applicable.

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
