# Peer review of "Biochemical and Initial Structural Characterization of the Monocot Chimeric Jacalin OsJAC1"

_ijms, 2021, doi:10.3390/ijms22115639_

Round 1
Reviewer 1 Report
minor
L24, 27, 346, and 347: carbohydrate-binding
Fig 6, 7, and 8: The symbol nomenclature of monosaccharides would be better to follow this link.
major
L 40-44: The relationship among JRL, DIR, and OsJAC1 is not familiar for general readers in the journal. For the readers, brief description about these proteins should describe based on the reference #8.
The description of the difference among OsJAC1, OsJAC1_P1, and P2 in Fig 2 can not see in the text. Please describe clearly and simply.
Fig 2: SEC analysis should perform by using haptenic saccharides in the buffer because the molecular mass of OsJAC1 (35,570), DIR (19,300), and JRL (18,390) (Table 1) do not match with the numbers.
In the case that the molecular mass of OsJAC1 JRL reduced to 30,000 and 5,000, respectively (Fig 2), readers confuse either these lectins bound to column resin via lectin-carbohydrate interactions, or these proteins formed tightly globules.
The paragraph from L111 to 137 including Table 2 may be difficult to have empathy for many readers. Please explain it well to be understood.
Because the 3-D structure of Jacalin is well understood, it will be available to indicate a reasonable explanation.
In Fig 4 and Table 3, a new parameter as JRL+DIR has suddenly appeared. The structure of the JRL+DIR should describe. Readers will confuse the protein as a mixture (right) or chimera (wrong).
In Fig 6 and Fig 7, it is difficult to understand why JRL+DIR and OsJAC1 (Fig 6) and JRL+DIR, DIR, and OsJAC1 (Fig 7) are separated in the different figures. Please explain simply.
In Fig 8, the numbers 85.2 oC and 59.1/85.1oC of the column OsJAC1 are separated. Please correct them.
The meanings of the colored tubes and cubes are unclear in addition to arrows and saccharides. Please improve the picture.
Please describe the method of size-exclusion chromatography in the Methods part.
Please add OsJAC_P1/P2 and DIR_P1/P2 in the list of abbreviations.
In the references, names of journals should write their abbreviations.
Author Response
Dear reviewer,
Thank you very much for your careful reading and the fruitful comments. As you will see, most of the comments have been addressed and as we think put a benefit to the former manuscript.
Thanks for your precious time,
Thomas Classen
Comments and Suggestions for Authors
minor
L24, 27, 346, and 347: carbohydrate-binding
Done
Fig 6, 7, and 8: The symbol nomenclature of monosaccharides would be better to follow this link.
Thank you very much for your suggestion. We hope that the color-coded structure representation is intuitive for a wider readership. However, if the editor insists that the generic form-based code shall be applied we are eager to change that in accordance.
Major
L 40-44: The relationship among JRL, DIR, and OsJAC1 is not familiar for general readers in the journal. For the readers, brief description about these proteins should describe based on the reference #8.
We added a cross reference towards table1 in L47 as description for the relationship of the two domains in the full-length protein. Table 1 contains a detailed range description in compliance with the used abbreviations. The text in line 44ff. has been slightly modified.
The description of the difference among OsJAC1, OsJAC1_P1, and P2 in Fig 2 cannot see in the text. Please describe clearly and simply.
Abbreviation for the main peak (OsJAC1_P1) and the secondary peak (OsJAC1_P2) has been added in L110 and L111.
Fig 2: SEC analysis should perform by using haptenic saccharides in the buffer because the molecular mass of OsJAC1 (35,570), DIR (19,300), and JRL (18,390) (Table 1) do not match with the numbers.
Similar experiments have been carried out using dynamic light scattering. These data are available in the SI and are discussed in the paragraph 126-141. The data are more detailed than the analysis by SEC. Or did we get the suggestion with haptenic saccharides wrong?
In the case that the molecular mass of OsJAC1 JRL reduced to 30,000 and 5,000, respectively (Fig 2), readers confuse either these lectins bound to column resin via lectin-carbohydrate interactions, or these proteins formed tightly globules.
We are aware that the SEC results could be misleading. For that reason, we provide an in-depth analysis of the results in the discussion (L332).
The paragraph from L111 to 137 including Table 2 may be difficult to have empathy for many readers. Please explain it well to be understood.
Because the 3-D structure of Jacalin is well understood, it will be available to indicate a reasonable explanation.
We are not sure what is meant with ‘reader’s empathy’. The respective paragraph contains the description of our findings using SEC and DLS, which we cannot change. Our attempts at explanation have been written down here, whereby we have tried not to formulate speculatively. The SEC data are further supported by dynamic light scattering data. These data are available in full in the SI. We would appreciate not getting into structural speculation here. Moreover, while 3D structures of related lectin domains have been published, these are unlikely to be helpful in explaining the self-association behaviour of the two-domain OsJAC1 protein, as revealed by DLS data.
In Fig 4 and Table 3, a new parameter as JRL+DIR has suddenly appeared. The structure of the JRL+DIR should describe. Readers will confuse the protein as a mixture (right) or chimera (wrong).
JRL+DIR as stoichiometric mixture of JRL and DIR domain was added as abbreviation in L247 and in the list of abbreviations (L537).
In Fig 6 and Fig 7, it is difficult to understand why JRL+DIR and OsJAC1 (Fig 6) and JRL+DIR, DIR, and OsJAC1 (Fig 7) are separated in the different figures. Please explain simply.
The focus were the two melting points of OsJAC1, which corresponds to the two domains.
In Fig 8, the numbers 85.2 oC and 59.1/85.1oC of the column OsJAC1 are separated. Please correct them.
The meanings of the colored tubes and cubes are unclear in addition to arrows and saccharides. Please improve the picture.
Thank you for your suggestions, we hope we adopted your comments accordingly into the figure.
Please describe the method of size-exclusion chromatography in the Methods part.
Description is provided in L464-469.
Please add OsJAC_P1/P2 and DIR_P1/P2 in the list of abbreviations.
Done
In the references, names of journals should write their abbreviations.
Done
Reviewer 2 Report
This is an interesting work dealing with the functional characterization of the DIR of a chimeric JRL from Oryza sativa OsJAC1. The authors demonstrate that the DIR domain display a carbohydrate-binding activity distinct from that occurring in the JRL.
However, results obtained with the recombinant JRL domain are lacking and should be included to make a valuable comparison with the recombinant DIR domain sugar-binding activity and look about the possible promiscuity of the DIR domain carbohydrate-binding activity.
Do the presented results mean that DIR domain is a lectin with a beta-barrel fold and a carbohydrate-binding specificity rather different from those observed in the JRL domain?
Author Response
Dear reviewer,
Thank you very much for your precious time and your valuable comments. We modified the manuscript and we are confident, that your review improved the publication.
Thanks for your precious time,
Thomas Classen
This is an interesting work dealing with the functional characterization of the DIR of a chimeric JRL from Oryza sativa OsJAC1. The authors demonstrate that the DIR domain display a carbohydrate-binding activity distinct from that occurring in the JRL.
However, results obtained with the recombinant JRL domain are lacking and should be included to make a valuable comparison with the recombinant DIR domain sugar-binding activity and look about the possible promiscuity of the DIR domain carbohydrate-binding activity.
The reviewer is probably referring to the results presented in Figure 6. Of course, we did attempt the same screening with the isolated JRL domain; as pointed out in lines 255-257, however, these data were not useful because inflection points were hard to discern.
Do the presented results mean that DIR domain is a lectin with a beta-barrel fold and a carbohydrate-binding specificity rather different from those observed in the JRL domain?
The current work was conducted to shed light on biochemical properties of the OsJAC1 protein and on “artificially” created recombinant proteins with its singe domains. As part of this study, we observed the so far unnoticed property of the respective DIR-protein to bind carbohydrates. This was unexpected and we thought to make this public available. In future experiments this effect will be investigated in more detail and this will then answer your question.
However, thank you very much for your comment. Further research needed to be conducted and is focus of future experiments in order to answer your question. Here we try to report the carbohydrate-binding activity of the DIR domain and want to raise awareness of its properties.
Round 2
Reviewer 1 Report
The authors have responded all to the questions by the reviewer.
Reviewer 2 Report
Although the authors did not answer the questions asked, I accept that this good quality article should be published in IJMS. However, I encourage the authors to further support their conclusions, particularly on the carbohydrate-binding activity of the DIR domain of OsJAC1.